# Productivity Improvement Using Simulated Value Stream Mapping: A Case Study of the Truck Manufacturing Industry

Fikile Poswa *, Olukorede Tijani Adenuga and Khumbulani Mpofu

Department of Industrial Engineering, Tshwane University of Technology, Pretoria-Campus, Pretoria 0183, South Africa
* Correspondence: fposwa44@gmail.com; Tel.: +27-737860726

**Abstract:** The accumulation of process waste in the production line causes fluctuations, bottlenecks, and increased inventory in workstations disrupting process flow. In this paper, the optimal process flow that will improve productivity using simulated value stream mapping (SVSM) for decision-making to provide consistency, minimise errors and non-value adding times in the implementation phase of VSM in the truck manufacturing industry. The proposed methodology applied a discrete event simulation for production process operations improvement to eliminate non-value adding times and provide good quality products at the lowest cost and highest efficiency. The results are the analysis of the current state of the production system in a South African truck manufacturing industry as a potential solution for the production system's future state. The identified non-value adding times in the six most critical workstations were eliminated by SVSM resulting in a productivity improvement of 4%, most importantly bringing the productivity to 95% and total cycle time improvement to 451 for small units and 466 for large units. The results proposed combined VSM and simulation techniques based on empirical data from the observation during time measurement. The Yamazumi confirms the issues observed and the NVA recorded by showing how close the process cycle times are to the TAKT time, which enhance the LEAN application by DES to increase productivity and performance improvement to remain competitive in the global economy.

**Keywords:** simulated value stream mapping; productivity improvement; lean enterprise; cyber-physical systems; spaghetti/yamazumi diagram



## 1. Introduction

Companies seek strong methodologies to increase productivity and performance improvement to remain competitive in the global economy. Process optimization focuses on quality, productivity improvement, and cost reduction, with most companies focusing on competitiveness and continuous improvement to achieve sets of key performance indicator (KPI) targets. Productivity is one of the key elements to measure how well the company is doing in the marketplace. Production flow is one of the factors taken into consideration when measuring and improving productivity. The flow is based on efficiencies and inefficiencies in the process where materials and tools are used in the production system [1]. This is based on the contributing factors that make the success of the applied tools [2]. New models are developed to ensure success and competitive advantages in the marketplace, which are complementary models to other continuous improvement tools and the existing paradigms in place to monitor the dynamics of the marketplace. These are cyber-physical systems (CPS) that support decision-making in the reconfiguration of the production system. These approaches emerged to support human machines in manual production systems. Lean manufacturing tools are used in the 21st century to ensure that companies perform at their maximum. The selection of ideal tools for continuous improvement and productivity improvement projects has been a challenge for SME automotive to date [3]. Repetitive activities are the most difficult to analyse because the disruption fluctuates and leads to big losses in time and productivity based on unforeseen factors. Most researchers ignore these

challenges because lean manufacturing failure is rarely reported, and yet they are required to serve as a guideline to a sustainable production line. The lack of a suitable framework on how to use lean manufacturing tools for the success of each project's continuous improvement support has been a major issue that is not addressed by researchers [4]. There are emerging technologies and collaborative robotics that are required to keep up with the pace of competitiveness for productivity improvement [5]. These technologies bring a sense of integrated monitoring and visualization to existing tools and techniques. The issues with these emergent technologies are the non-empirical evidence to show the effectiveness of these approaches. Lean manufacturing has played a huge role in making sure that all levels of production and manufacturing can deliver their optimal processes. Lean manufacturing is the tool that integrates production processes, tools, materials, and operators to achieve the common objective of being efficient. Mathematical models, laws, and algorithms are mostly neglected in the production environment, yet they are valuable tools for analyses when statistical process controls are applied. Efficiency is one of the most important aspects of productivity, which needs to keep it at its maximum all the time. Value Stream Mapping has been an effective tool in recent years to visualize the production and manufacturing system [5]. Value Stream Mapping (current analysis and optimum solution) and Anylogic agent-based simulations were used to examine the manufacturing process parameters and apply lean principles. This led to an evaluation of the lean performances in terms of key performance metrics for the goal VSM. Utilizing the data provided by Value Stream Mapping and having the option to select the optimal scenario out of several options are two advantages of including agent-based simulation in the design and analysis of the value flow in the production chain. It results in significant time and cost savings without adding to resource waste [6]. This shows how non-value adding operations disrupt most of the production lines mostly in the automotive manufacturing system because non-value adding is considered as value adding times [7]. Productivity improvement happens when there are systems that ensure that all the KPIs involved are monitored to fulfil all the demands. These KPIs and demands refer to quality, efficiency, and low cost. This article aims to simulate value stream mapping (SVSM) as a lean assessment tool for decision-making in the continuous improvement process to provide consistency, minimise errors and non-value adding times in the implementation phase of VSM in the truck manufacturing industry. Section 1 discussed the review of the current production state challenges and potential solutions for the future state of the production system. Section 2 discussed the literature reviews on how the production system behaves, and how the methods are applied in production to seek improvement opportunities. The methodology in Section 3 discusses how conventional value stream mapping (VSM) is complemented by simulations where applied data are ideal to describe the evidence of complementary time study data method, and data analysis by VSM and SVSM schematic diagrams. The data were simulated using AnyLogic Discrete Event Simulation (DES) and manual VSM parameters. Section 4 presents the results that show the study application was to improve the productivity of the truck manufacturing industry, while the last section presents the conclusion based on empirical data from the observation during time measurement. The Yamazumi confirms the issues observed and the NVA recorded by showing how close the process cycle times are to the TAKT time, which enhance the LEAN application by DES to increase productivity and performance improvement to remain competitive in the global economy.

## 2. Literature Review

The lean manufacturing system has been a driving force for productivity improvement in the automotive industry, which extends from the long-time leader, the Toyota Production System (TPS), known for changing how things are done in the automotive industry. To be competitive in a sustainable environment, different techniques are used to eliminate waste that results in creating non-value-adding times in the production system [8]. These non-value-adding times contributed to a more complex process when the classification of the process is not done properly because the nature and dynamics of the process are not visible

enough [9]. Value Stream Mapping is the commonly used tool in waste reduction projects, where it provides the visualization of the entire production system. This helps to identify waste and classification done based on the nature of the waste for well-defined decision-making. VSM is a tool that can easily be used with other waste reduction techniques to enhance core functionality that supports using VSM as a waste reduction tool and proper strategies on how to reduce the time wasted on the system [10]. Non-value adding is the time wasted by not adding value to the product, which translated to not adding any value to the customer because the customer is not willing to pay for that time. Despite the VSM capabilities, there are shortcomings when the production system shows the characteristics of being dynamic. VSM can be complemented using digitalization to bring flexibility that will overcome these shortcomings [11]. Though digitalization has different elements and applications that might be costly when applied to enhance VSM, simulation has been around for quite some time, and there is a new application designed to complement VSM. VSM is a lean manufacturing technique that aligns with other tools used for process optimization and continuous improvement projects. This application paves a way to apply those techniques in a way that will be less costly and more effective. The implementation of lean manufacturing is more expensive when everything must undergo a trial or assurance before it can be up and running, and this is a time constraint to a project [12]. Time consumption happens when appropriate lean manufacturing tools are not used properly, which happens when the uncertainties of the process have not been identified properly. Identifying the cause of these uncertainties become difficult when the production system has the characteristics of the dynamic production system.

Simulation in a production environment provides stability by assuring that real-time use of simulation identifies the hidden cause of uncertainties in decision-making [13]. Simulation in business practice assures the production system. Performance and high expectations are the key drivers of a successful production system [14]. This drives the determination of seeking a production system that continuously improves and aims to use lean as the assessment tool [15]. The use of lean manufacturing and simulation models is to effectively apply a business strategy to be competitive and efficient [16]. Most automotive industries use discrete manufacturing systems where queuing takes place, causing delays in the process workstations. Too many delays increase waiting time. In a production environment, waiting times are recorded as the idle time that a customer is unwilling to pay for, thereby decreasing productivity. Insufficient process reliability, supply chain, and an inactive workforce cause idle time in the system. In this kind of manufacturing system, some disruptions are caused by dynamics in the production system, which can be evaluated using Discrete Event Simulation (DES) [17]. The simulation provides management with different alternatives compared and trailed for better decision-making in the application of lean manufacturing. The application of simulation models can minimize these challenges by analysing these complex structures [18]. The simulation is a good tool to assess a production system without any physical intervention in the system. This approach gives an overview of the system compared to manual operator intervention. The emerging use of information communication technology (ICT) and information technology (IT) in the manufacturing sector has increased information transparency across all functional areas of the organization. This transparency helps improve the supply chain and just-in-time (JIT), production deliveries, minimizing the dynamic disruption and uncertainty in the production processes. Proper alignment of every aspect that contributes to production improves efficiencies, thus improving productivity [19]. The industrial revolution shifted the emphasis from a production system that focuses on quality to one that focuses on both quality and production efficiency. This forces business efficiency to pull in the direction of global competitiveness by using advanced technological innovation to improve productivity [20].

The application of cyber-physical systems (CPS) in manufacturing systems has improved production systems and enhanced the use of digital simulations to improve real times and cost-effectiveness in the production system. A simulation-based approach in

a production system has brought about significant advancement, especially in processes involving human manpower [21]. Production visibility is essential in production improvement because it brings about the overall flexibility of the production system. SMEs in the automotive industry apply lean manufacturing in an advanced manner by keeping its original principle but applying a computer-based simulation to optimize the production process. Lean manufacturing is famous for production system improvements, but it is difficult for dynamic processes where visibility is critical. A simulation is a complementary tool for the difficulties of production process dynamics [22]. Evaluating the environmental impact of a dynamic production process needs more visibility to support decision-making. VSM is a reliable tool in process visibility, making it the most used tool in lean thinking and implementation. It is a vital tool in developing knowledge and understanding of the production value. Simulation complements VSM by utilizing and analysing the data collected from dynamic processes [8]. It is a good practice in discrete sequence production systems where resources such as facility layout, tool location, and part supply set the pace from the warehouse and process complexity, rather than the machine setting the pace. Processes are executed when the production system has an adequate facility layout to allow the process to be performed at the maximum best [23–25]. VSM can only analyse one product at a time, which is different from simulation models. Although VSM visualizes the process by showing the cycle and changeover times, it fails when the process is dynamic. Mixed production systems cannot be studied with VSM alone; multiple projects would have to be implemented to accommodate all the models/products produced in that system [26–28]. Value stream mapping visualizes the data collected from the process by plotting each component on the map. VSM is bounded by symbols used to represent the components in the process; however, some information is not displayed in the VSM map to visualize how the process is doing. VSM is a tool that supports the improvement process analysis in the manufacturing system. The results displayed by VSM are of great use when line balancing is applied to validate how each process contributes to process reliability and capability. A reliable process improvement is based on the characteristics required to perfect multiple activities and provide flexibility where necessary [29,30].

## 3. Materials and Methods

### 3.1. Materials

The use of the quantitative research approach in this paper was to determine the relationship between the processes in the production system for SMEs in the automotive industry. Quantitative research is a descriptive approach, based on time measurement and was used to gather data on time measurement in all stations involved in the truck production system. Secondary data for the time study was obtained from a South African truck manufacturing company. The data applied in this paper are used to determine the effectiveness of the applied method, and the simulated value stream mapping (SVSM) was employed to describe the effective use of lean manufacturing in the production system. This is to show that conventional tools in the dynamic system need to be complemented by ideal models such as simulations. In this paper, discrete event simulation (DES) is a complementary modelling approach used to describe these discrepancies in the traditional value stream mapping.

Current State VSM

Value stream mapping is constructed using the cycle time, change over time, operators, and process description. This process visualizes the production system and the behaviour of each workstation. This populates the waste recorded while collecting data during observation of the current state of VSM. All elements are exhibited to point out which areas have problems, and potential solutions are developed based on what is visible. To validate the populated and visualized concerns, the cycle times are categorized as VA and NVA to show how much the populated concern affects the process. The study of the NVA contributed to the cycle time envisaged as the situation and reckoning with the presence

of one of the seven waste leans. The contribution of the VA and NVA is shown in the Yamazumi diagram in Figure 1, which depicts how the processes are responding to the TAKT time. This is based on empirical data from the observation during time measurement. The Yamazumi confirms the issues observed and the NVA recorded by showing how close the process cycle times are to the TAKT time

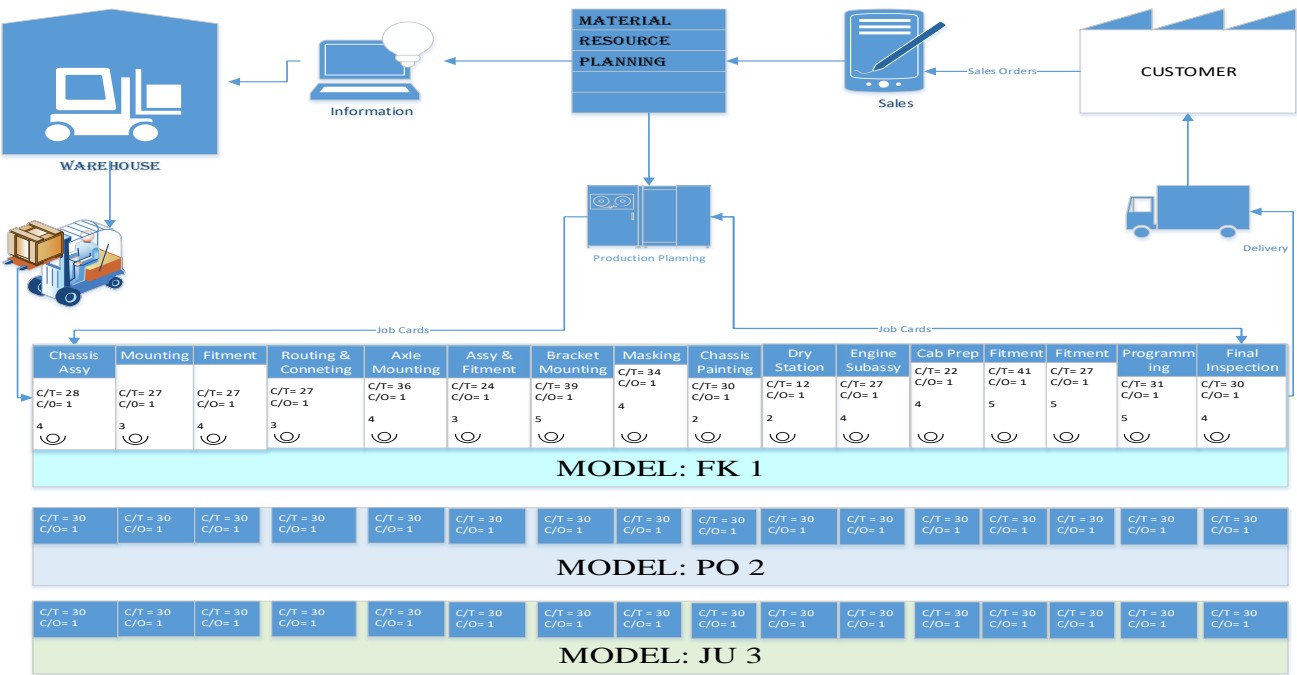

**Figure 1.** VSM modelling of the current state of the truck manufacturing industry (FK1, PO2 and JU3 are named after the workstations; C/T is cycle times and C/O operations times).

The current state of VSM is the logical representation of how the production system behaves, where the VSM schematic diagram is filled with times that coordinate the production system as shown in Figure 1. When the current state is completely plotted, all the measurements are in place to do the analyses, the visualization of every aspect of the production system is clear, and the waste can be identified. The logical diagram of VSM shows how the waste is accumulated in the production system, and suggestions for improvement are observed and proposed based on this logical presentation. VSM visualizes the process waste, mainly focusing on times that do not add value to the system output. This is measured by process times and the number of units produced in the time allocated for the shift. The VSM used in this research study is designed from Microsoft Visio because the software is efficient and easily transferable to any document format.

### 3.2. Methods

The link between research design, question and hypothesis was investigated, considering how the parameters were set up to establish the current state of VSM, and the current state of SVSM. These parameters define the data collection using the time study method, and data analysis by VSM and SVSM schematic diagrams. The data were simulated using AnyLogic Discrete Event Simulation (DES) and manual VSM parameters. System design and process optimization were done to improve productivity and eliminate hidden waste. The choice of a discrete event simulation for this research is basically to provide a logic system in discrete processes for autonomous and continuous times. The work measurement technique used in this study is determined by the time series involved in completing one unit in the production line and to compare different cycle times that determine the value in the flow and reliability of the production process.

VSM application in the truck manufacturing industry is an improvement strategy used to improve process efficiency. An effective production system has balanced cycle times in good productivity and efficiency. The simulation process was used as an imitation of the processes in practice and considered as an applied mathematics technique to solve VSM techniques, which have been integrated to optimize the production process and minimize errors made in lean manufacturing application projects. VSM was used to visualize what has been fed to the technique, which is different from simulation because the results cannot be manipulated compared to VSM. VSM was used to identify all the types of waste involved and the categories in which they belong in the value chain of the truck manufacturing industry. When combined, these two techniques bring more attention to detail by complementing errors that might occur as a result of the project, which is very costly in many cases.

It brings the analysis of activities and discrepancies identified when a time study was conducted. Identifying VA and NVA times helped the process to minimize or eliminate those tasks and times using time and motion study to organize how the labour was aligned to the daily demand of the production; how actual cycle times perform against the production TAKT time as presented in the following equation:

$$T_c = Tr + \sum Tsi \tag{1}$$

In evaluating the work assignment

$$T_c = T_s + T_r \tag{2}$$

Production line efficiency is determined as

$$E = \frac{T_s + T_r}{\frac{60}{R_p}} \tag{3}$$

While production line efficiency balancing

$$E_b = \frac{T_{WC}}{wT_s} \tag{4}$$

Determination of the daily rate of the production system

$$R_p = \frac{D_a}{S_d H_{sh}} \tag{5}$$

$$T_p = \frac{60}{R_p} \tag{6}$$

Production work content is determined as

$$W_C = \sum_{k=1}^{n} T_c \tag{7}$$

$$w = \text{Minimum integer} \geq \frac{T_{WC}}{T_s E_b} = \frac{T_{\frac{T_{WC}}{T_s E_b}}}{T_s \frac{T_{WC}}{wT_s}} \tag{8}$$

Distribution Factor to the Assembly Line

Part arrivals are based on exponential distribution, which describes the process of occurrence in discrete events. This shows the number of parts and chassis coming through the assembly line. It is one of the findings identified in the model because it causes congestion in the system by supplying too many parts. This tends to overload other

stations with less capacity in resources. Poisson distribution application was adopted and is defined by Equation 9:

$$f(x) = \frac{u^x e^{-u}}{x!} \tag{9}$$

The secondary data was coded to create the value stream mapping the current state and input data into AnyLogic discrete event simulation. As part of DES analysis, entities need to have sequential events that are timeous and behavioural. The time series in the case study populates discrete events occurring in the production line and other departments contributing to making production a success. This paper focuses on VA and NVA times, achieved by looking at all the aspects that contribute to the working states and coordinates of these times. Total cycle time is the result of VA and NVA combined and a representative of how the operation is running in line with process control parameters. The process control parameter used in this paper is the TAKT time, as a pacesetter to all processes in the production line. These parameters aligned the process with the schedule as the final output of the production line. It serves as a guideline for the number of resources required to complete one unit; thus, it helps in the scheduling of the expected output. It is a tool that prevents spiking process fluctuations by ensuring each process is done under TAKT time. The work measurement technique identifies the time wasted in the process by not adding value to the customer demand (NVA). The observation done while measuring the cycle times is recorded and used during the analysis phase. This is a major factor in identifying how these wastes accumulate and allows classifying the type of waste. Figure 2 presents the line balancing for the production line balancing of the current state of the truck manufacturing industry, while Table 1 is the plotted truck manufacturing assembly's current state to visualize the current situation.

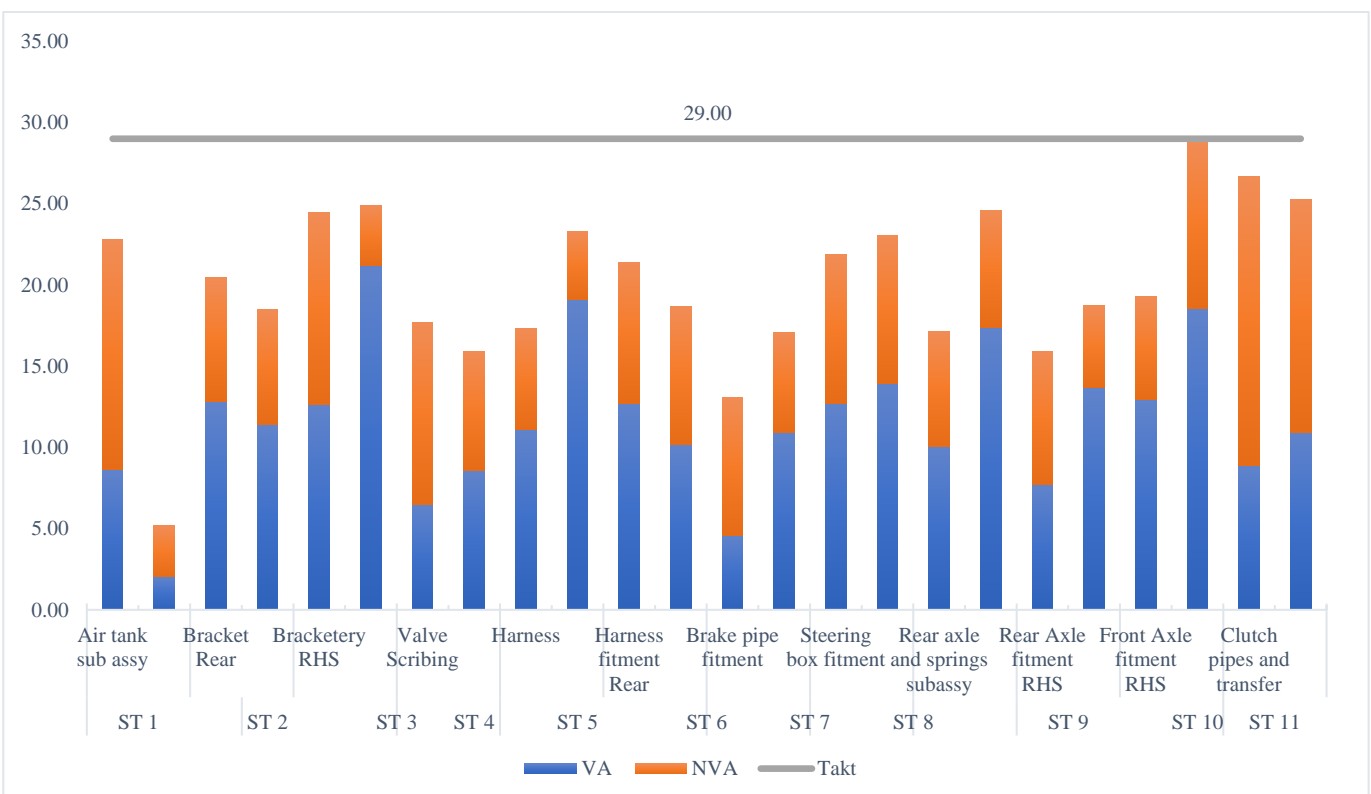

**Figure 2.** Initial state of the production line balancing of truck manufacturing industry (VA—value added times; NVA—Non-value-added times; Takt time-metric for production time determination).

The flow in the production system is one of the important aspects of process improvement, which shows how VSM can be a valuable tool for analysis. TAKT time is one of the

coordinates controlling how the production system should operate effectively (Equation (9)) as a guideline to the determination of cycle times of each process to avoid abnormal fluctuation in production processes. Tables 2 and 3 present the requirements of the production system, and the times in Table 1 are the data derived while executing these targets.

The following are the representation for the abbreviations.

CA—chassis assembly, assembly of suspension legs; PE—peripheral mounting, stabilizer bracket mounting, platform supporter mounting, bogey assembly; FS—fitment of the steering box, power steering pipe, brake valves, air tanks, and compressor pipe connecting; ROU—routing and connection of air pipes; R&F—Rear and front spring mounting, tag axle and front axle mounting; ASS—assembly of the rear axle, fitment of rear axle and prop shaft; ASL—assembly of LSV rods and mounting into the rear axle, tail light and wheel choke brackets, battery box, fuel tank bracket mounting, cab tilt mechanism mounting; CM—chassis masking, loom connection, chassis rubbing, fitment of booster pipe and speed sensor, isolator board mounting; CP—chassis painting; DS—drying station; EAS—engine subassembly, engine mounting, and preparation of cooling pack. Fitment of exhaust and gearbox; CAB—cab preparation and drop, fitment of soundproof, drag link prep and mounting, air cleaner assembly, mirrors and mudguard mounting; FF—fitment of the fuel tank, tyres, and spare wheel; BHB—bumper, headlight, battery box, trailer loom and mirror, rear mudguard fitment, and brake test; PFV—programming, filling with fuel, vehicle inspection brake roller, mechanical inspection and start-up; VFQ—vehicle final quality inspection.

**Table 1.** The truck manufacturing assembly's current state is plotted to visualize the current situation.

| Station | Process | No. of Op | FKI | NVA | VA | PO2 | NVA | VA | JU3 | NVA | VA |
|---|---|---|---|---|---|---|---|---|---|---|---|
| 1 | CA | 4 | 28 | 10.08 | 17.92 | 30 | 10.8 | 19.2 | 24 | 8.64 | 15.36 |
| 2 | PE | 3 | 27 | 12.15 | 14.85 | 23 | 10.35 | 12.65 | 28 | 12.6 | 15.4 |
| 3 | FS | 4 | 27 | 10.8 | 16.2 | 32 | 12.8 | 19.2 | 37 | 14.8 | 22.2 |
| 4 | ROU | 3 | 27 | 11.34 | 15.66 | 23 | 9.66 | 13.34 | 30 | 12.6 | 17.4 |
| 5 | R&F | 4 | 36 | 13.32 | 22.68 | 25 | 9.25 | 15.74 | 31 | 11.47 | 19.53 |
| 6 | ASS | 3 | 24 | 10.56 | 13.44 | 18 | 7.92 | 10.08 | 23 | 10.12 | 12.88 |
| 7 | ASL | 5 | 39 | 14.04 | 24.96 | 32 | 11.52 | 20.48 | 31 | 11.16 | 19.84 |
| 8 | CM | 4 | 34 | 15.64 | 18.36 | 32 | 14.72 | 17.28 | 29 | 13.34 | 15.66 |
| 9 | CP | 4 | 30 | 15 | 15 | 34 | 17 | 17 | 34 | 17 | 17 |
| 10 | DS | 2 | 12 | 9.12 | 2.88 | 12 | 9.12 | 2.88 | 12 | 9.12 | 2.88 |
| 11 | EAS | 3 | 27 | 10.26 | 16.74 | 29 | 11.02 | 17.98 | 18 | 6.84 | 11.6 |
| 12 | CAB | 4 | 46 | 14.26 | 31.74 | 22 | 6.82 | 15.18 | 33 | 10.23 | 22.77 |
| 13 | FF | 5 | 41 | 9.43 | 31.57 | 41 | 9.43 | 31.49 | 41 | 9.43 | 31.57 |
| 14 | BHB | 6 | 27 | 7.02 | 19.98 | 67 | 17.42 | 49.58 | 47 | 12.22 | 34.78 |
| 15 | PFV | 3 | 31 | 17.05 | 13.95 | 21 | 11.55 | 9.45 | 24 | 13.2 | 10.8 |
| 16 | VFQ | 4 | 30 | 6.3 | 23.7 | 30 | 6.3 | 23.7 | 30 | 6.3 | 23.7 |

**Table 2.** The initial production target of the production system.

| Schedule Target | Available Time (min) | TAKT Time (min) |
|---|---|---|
| Daily | 9 | 48.3 |
| Weekly | 45 | 48.3 |
| Monthly | 180 | 48.3 |

**Table 3.** Initial productivity model input using VSM.

| Mixed Models | Total Cycle Time (Minutes) | Total Throughput/Day | Monthly Throughput | Monthly Target |
|---|---|---|---|---|
| FKI | 486 | 8.05 | 161 | 180 |
| PO2 | 471 | 8.3 | 166 | 180 |
| JU3 | 472 | 8.29 | 165.8 | 180 |

The Yamazumi diagram was developed to determine the current state as a tool used to do line balancing based on the characteristics of the VSM and DES process evaluation. The individual work tasks of the two process elements detailing the time taken are the critical characteristics required for productivity improvement. The importance of TAKT time in the process is to stabilize the process and provide daily information for production throughput. It is a good tool for a production-based process on what is planned for the day. Having TAKT time in place for all the products will determine the response to customer demand. The process flow in VSM seems to flow smoothly because everything is within the TAKT time. When everything falls within the TAKT time, it is regarded as perfect execution in the manufacturing industry. In lean manufacturing, TAKT time is used to test a proposed pace to improve productivity. It also acts as a boundary to determine the process flow to ensure no disruptions in the process. It also monitors fluctuation in the process and becomes a signal when these fluctuations exceed the boundary (TAKT time) set to meet customer demand. This technique is used in VSM mainly and is different for each unit produced, depending on size and type. The Yamazumi diagram presented in Table 1 shows the current situation for the truck manufacturing assembly, where all the parameters used in this study are plotted to visualize the current situation. Orders come from the customer from the sales department and are shared with the planning department. The information is evaluated and filtered based on the available material stock, and the information is shared with different stakeholders to prepare to produce the product. All the processes involved are prepared; accordingly, the flow of material starts from planning by distributing information.

## 4. Results

### 4.1. Future State VSM—Effective Production Process to Improve Productivity

Process modelling using DES shows the importance of keeping track in series rather than segregating the workstation. Isolating the workstation has a negative impact on production flow. The changes made to improve a problematic workstation or a workstation that can be improved for better flow affect the performance of the next workstation. This causes fluctuations in the production process and delays, resulting in poor quality and customer service. When looking at the results of VSM compared to SVSM, it shows the importance of the value chain. In SVSM, there are no costly delays when testing the improvement of a change made in one variable; the impact can be seen immediately without spending more time and money on the production lines. Failures are seen, observed, and rectified before the actual piloting of the improvement project. In production processes, workload distribution is essential to minimize bottlenecks, fatigue, and job complexity. Figure 3 illustrates an emphasis on how unforeseen waste can be hidden in traditional VSM. A well-distributed production process flow improves efficiency and productivity. This helps identify and eliminate non-value-added operations in the process where flow and value chain are key performance indicators. It is determined by the cycle times, which show the adherence of the process sequence relative to the TAKT time of each model. Production processes are dependent on the logistics supply/delivery factor. When there is poor logistics delivery leading to low productivity and if production processes are not aligned to supply chain processes, production will encounter problems relating to parts and materials lying around the production plant congesting the operator's movement, which will lead to large non-value adding times. South African automotive industries mostly find themselves in this situation due to a lack of knowledge on how to conduct their business by practicing lean manufacturing early. The practitioners tend to focus on production improvement projects that lack track and mostly fail in execution, resulting in value chain issues. Simulation enhances accuracy when properly applied to minimize errors that may occur during the execution plan. As illustrated in Figure 3, the application of the two paradigms shows that the VSM is ideal for non-complex systems, and simulation simplifies complex production systems. In manufacturing processes, it is common that companies want to maintain a free flow of processes based on easy management when

all processes are in series. It is good practice that subassemblies and offline processes are used to minimize delay in the production line, which brings more relief and reduces the complexity of the system. In the system used in this research project, Figure 1 shows that processes such as spray painting, drying, programming, and vehicle quality testing can be taken offline. This will allow free flow in the production process and minimize time spent waiting for these processes because they cannot be interrupted.

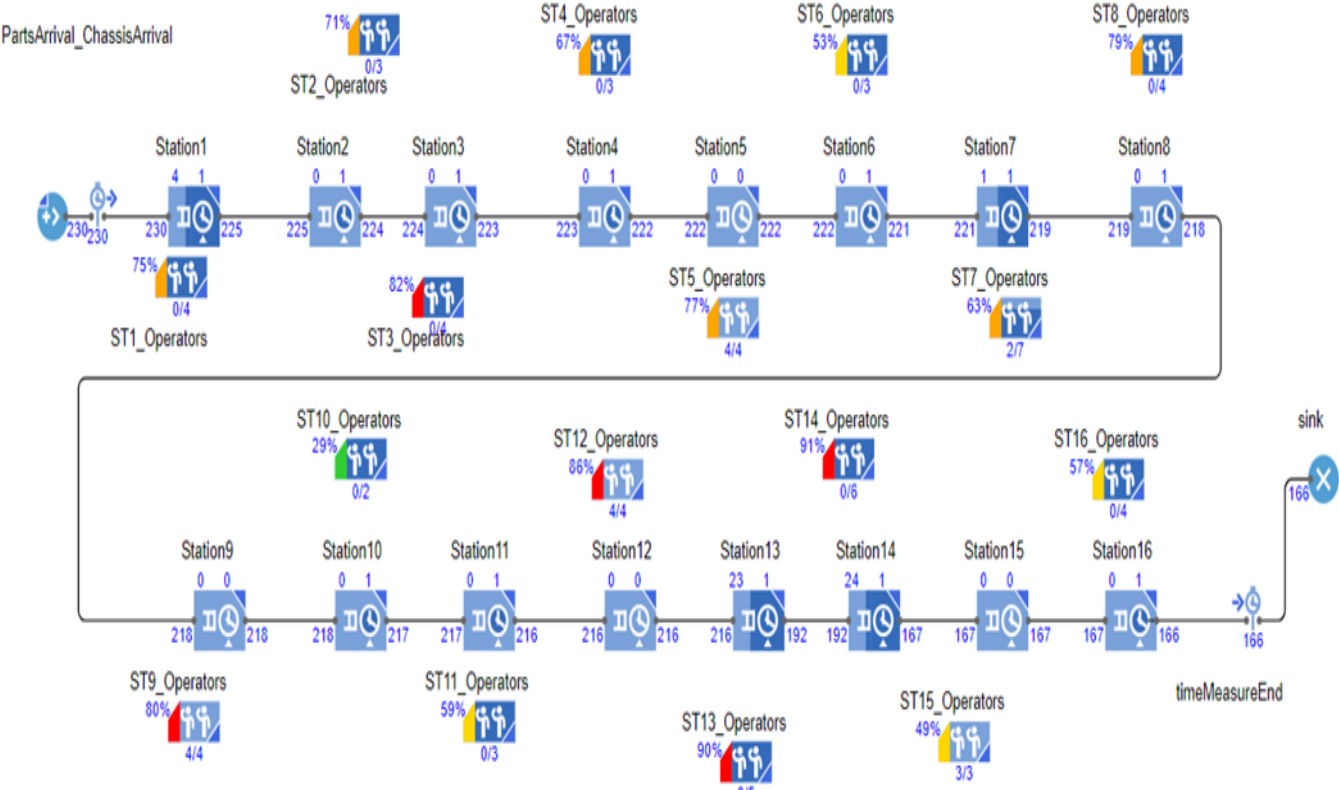

**Figure 3.** Simulated production system for the truck manufacturing industry.

### 4.2. Effective Application of Lean Manufacturing in the Automotive Industry

Companies, mainly SMEs, always try to improve their economic impact in the automotive industry by assuring their support to the industry through on-time delivery, high quality, and better service provision. These processes are driven by an efficient company process supported at all levels of the organization. Lean manufacturing has played a significant role ever since it was outlined as a way of eliminating waste and providing stability, efficiency, and an overview of where to improve for better production of products for customers. The effectiveness of lean manufacturing does not rely on the results generated; it depends on how the process has improved, and the key elements that are needed to ensure the non-re-occurrence of similar constraints and waste. The application of lean manufacturing principles has shown that all the stations in the process must be aligned to add value to the process chain of the manufacturing system in the automotive industry. Failure to adhere to these principles leads to major time loss due to breakdowns and other factors contributing to production processes. Stations 8, 9, and 10 (chassis painting and drying) can be optimized by completely removing them from the assembly line. The processes conducted in these workstations can cause significant constraints when they have breakdowns that affect the whole production line. One hour lost repairing and maintaining these workstations can result in approximately 14 h of production downtime because each workstation loses the same amount of production time or more than the repaired workstation. Figure 4 shows an improved VSM called the future state, which is improved from the current state of the production system.

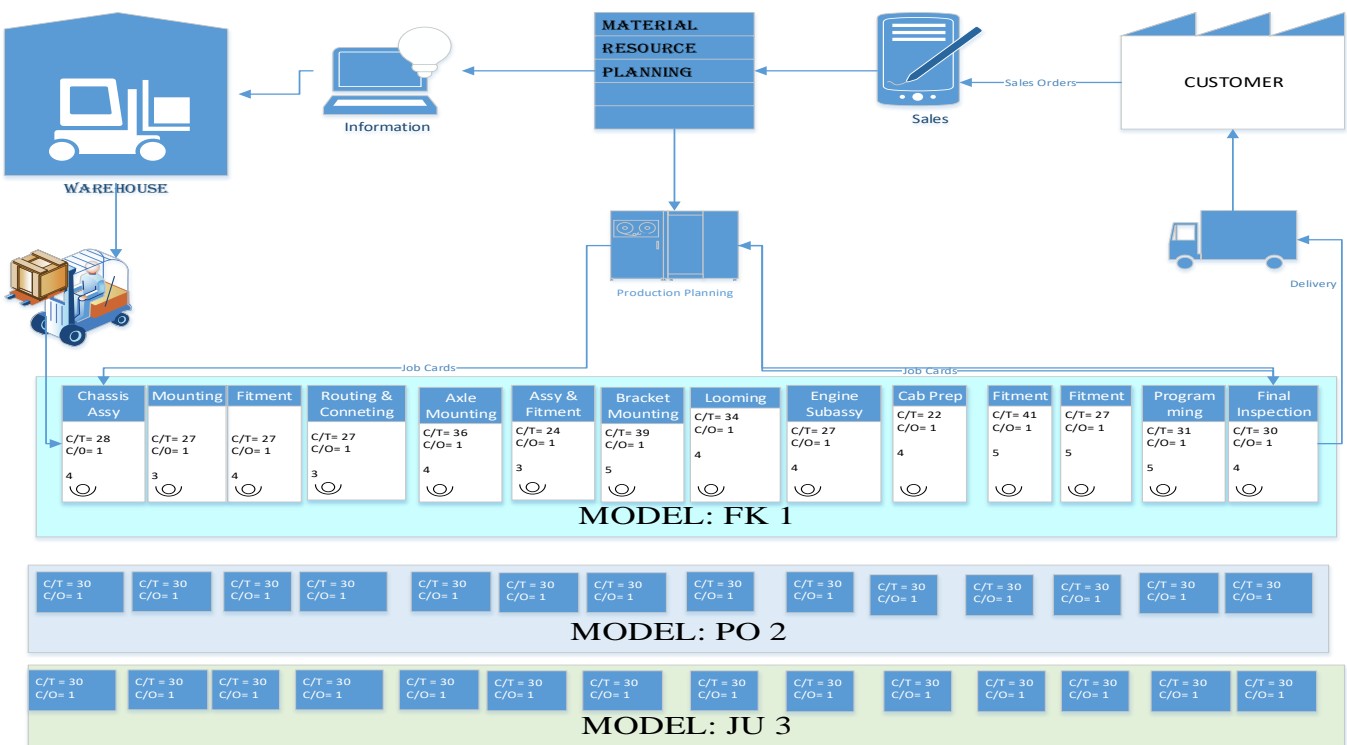

**Figure 4.** VSM modelling of the future state for the truck manufacturing industry (FK1, PO2 and JU3 are named after the workstations; C/T is cycle times and C/O operations times).

Value stream mapping is constructed using the cycle time, change over time, operators, and process description. This process visualizes the production system and the behaviour of each workstation. This populates the waste recorded while collecting data during observation of the current state of VSM. All elements are exhibited to point out which areas have problems, and potential solutions are developed based on what is visible.

Figure 4 presents the setup that allows flexibility in the process by enabling all the models with a smaller capacity than the ones studied to be manufactured as a good technique in support of mass customization. TAKT time is not used in simulation because it increases the delay time in process execution. The production system simulated and studied demonstrated that process balancing was not practiced well because some workstations are more overloaded than others is a common practice in the automotive industry because of product improvement.

## 5. Systematic Analysis of the Production System

### 5.1. Productivity Improvement through SVSM

Productivity improvement is a key strategy to maintain and gain a competitive edge in the automotive industry. The current process results clearly show that the overall production system is not compatible to yield the desired results. As shown in Figure 5, the targets are normal and can be achieved through a well-established production system where the processes are operating at their optimal.

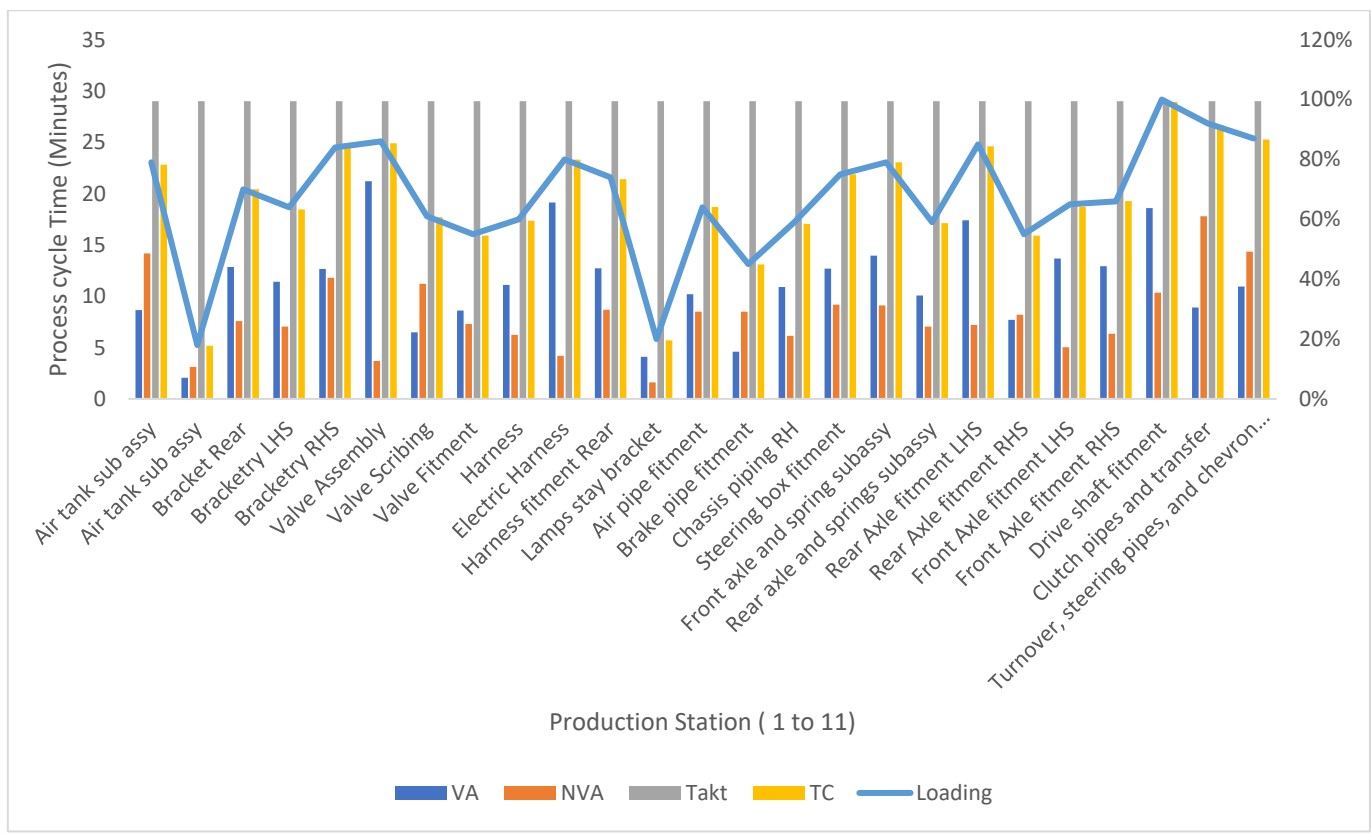

**Figure 5.** Production station against production cycle time (VA—value added times; NVA—Non-value-added times; Takt time-metric for production time determination).

Stations 1, 2, 3, 7, 8, 10, and 11 are good examples of this practice, as shown in Figure 5. The application of lean manufacturing principles has shown that all the stations in the process must be aligned to add value to the process chain of the manufacturing system in the automotive industry. Failure to adhere to these principles leads to major time loss due to breakdowns and other factors contributing to production processes. Some of the workstations can be optimized by completely removing them from the assembly line and having subassemblies that increase availability. The processes conducted in these workstations can cause significant constraints when they have breakdowns that affect the whole production line. One hour lost repairing and maintaining these workstations can result in approximately 14 h of production downtime because each workstation loses the same amount of production time or more than the repaired workstation.

*5.2. Results of Productivity Improvement through SVSM*

Productivity improvement is a key strategy to maintain and gain a competitive edge in the automotive industry. The current process results clearly show that the overall production system is not compatible to yield the desired results. As shown in Figure 5, the targets are normal and can be achieved through a well-established production system where the processes are operating at their optimal.

The value stream mapping shows an overview of the production system, using the information gathered during data collection to understand the production system's behaviour. This proves that non-value adding time is the cause of inefficiency. This shows how the workstations are loaded and the number of processes done in one workstation. That shows the effectiveness due to congestion because of these processes. This shows how tools, warehouse distribution, manpower utilization, and the facility is arranged to allow optimal performance of production processes. Production systems are seen from a different perspective when it comes to the utilization of resources, which provides the aspect of most

companies practicing the replenishment of stock. Productivity and efficiency are affected negatively by this practice, where quality is being compromised because the targets are not met. Figure 6 shows that the processes are not operating at the best level based on the results below. The are several factors affecting the production system not to achieve its desired productivity. The potential factors that might affect this production system are highlighted by just looking at the NVA, though it is understandable that some of the NVA is necessary. Proper execution of the processes defined by the cycle time determines productivity and raises flags on how the resources are utilized. In most companies, these flags are ignored if the production runs without any disruptions shown, yet this contradicts the theory of constraints, which states how small disruptions can contribute to the loss of productivity. The TAKT time in each workstation is the same, keeping production flow under control. Allowing fluctuating cycle time because the nature of the operation involves humankind that works at a different rate. However, the time lost in each workstation is equivalent throughout the system.

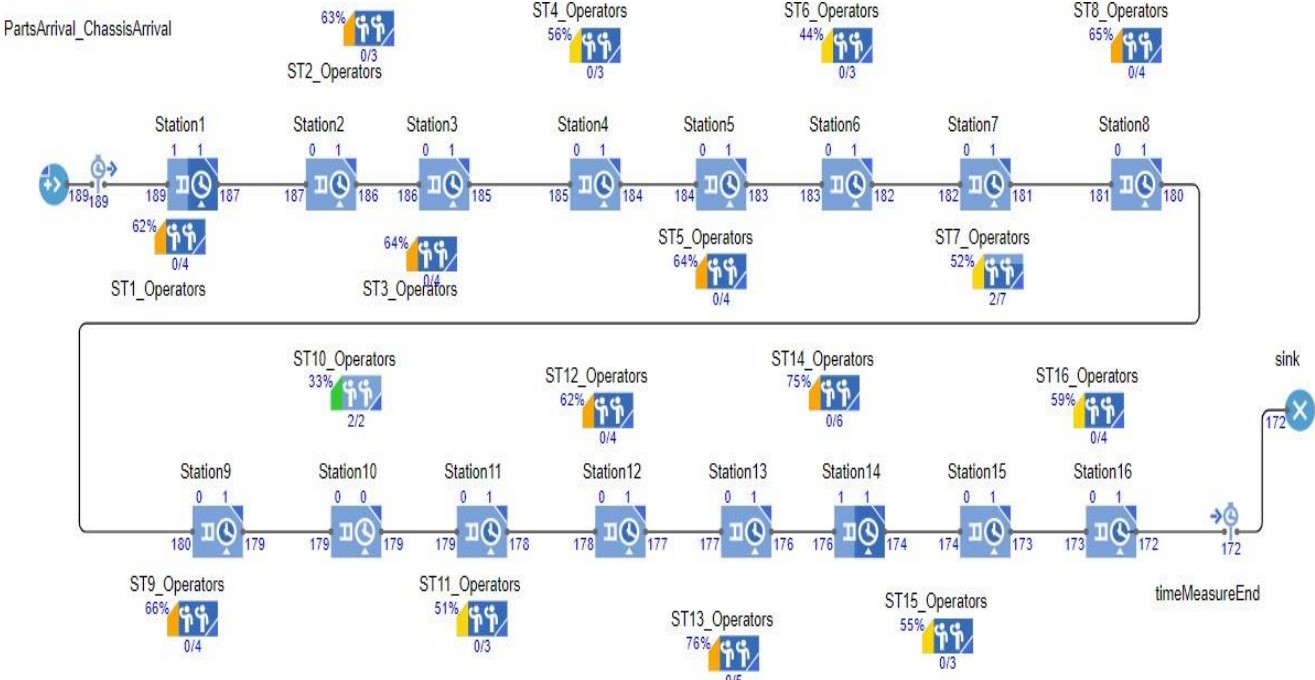

**Figure 6.** VSM of the productivity improvement through SVSM.

The case study shows that some of the processes are overloaded because they operate closer to the TAKT time. When the processes are operating too close to the TAKT time, they are putting the system into a pressured operation. Even distribution of process operation balances the system and improves productivity. This is caused by high variation in processes, and the production rate is determined by the capacity of the resources. After simulating the production system using DES, it shows that the system is balanced, see Figure 7, yet it does not meet the target. This is where we revert to issues that were highlighted during the study, to know the gaps and the potential solutions.

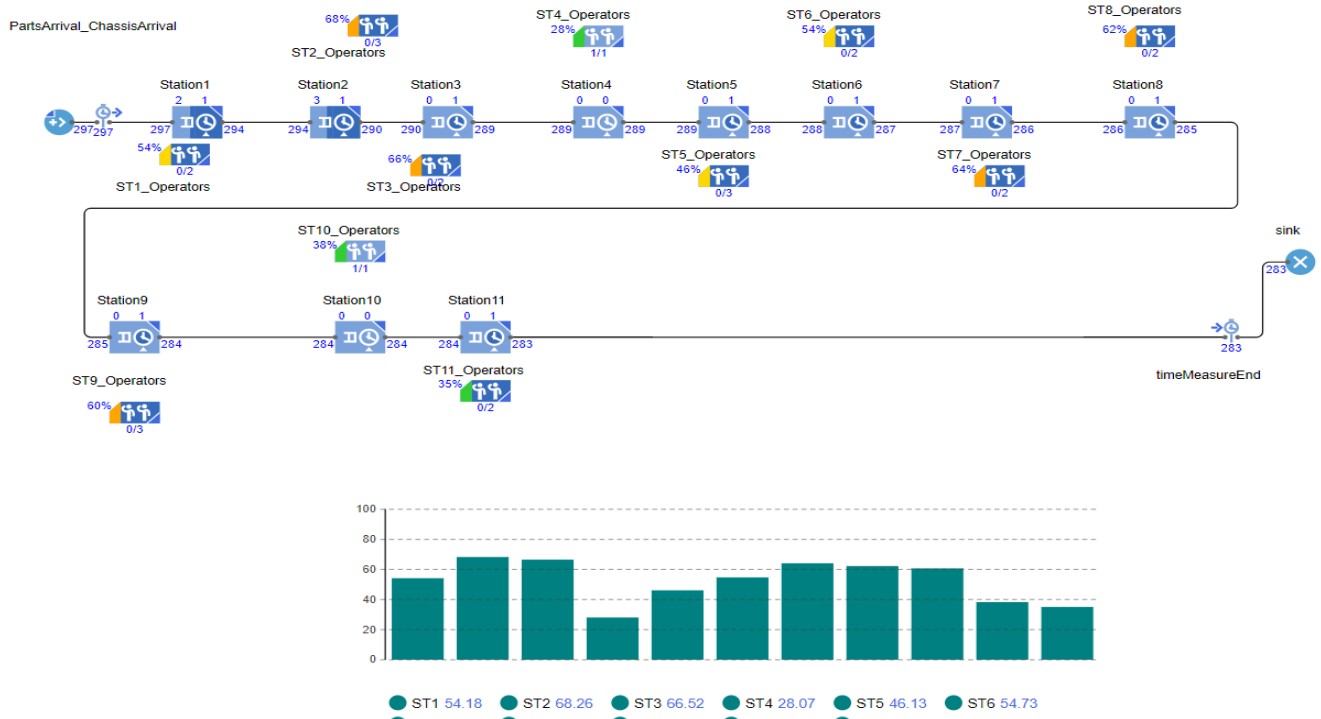

**Figure 7.** Balanced production system using DES.

The lean assessment tool intends to improve the performance of the overall production system where the value chain is the key element. The flow determines how processes can meet customer demand with more emphasis on material suppliers and available space. This supports the continuous flow of the process with fewer or no constraints that cause uncertainty in process execution. The use of technology in the production system brings more stability and shows how and where improvements are needed. Simulation is a good complementary tool that needs to be used as an assessment and a validation tool for lean manufacturing projects.

*5.3. Material Handling Support to Improve Productivity*

The effective application of lean manufacturing improves material handling and safety in the workplace. This happens through the improvement of material and parts movement, packaging and storage. Material handling and safety are crucial elements of any process because of their vital contribution to productivity and quality. These are determined by improving the distance covered to collect the part, the safety of the surroundings and how the material is placed to support a safe working environment. Figure 8 shows how the movement, material handling, and parts storage around the workstation is visualized and improved using lean manufacturing. The effective application of this technique minimizes unnecessary movement and gives an overview of the distance covered for each process. It identifies the longest paths, double handling of the process and shows the non-value adding times. Constraints are identified and dealt with to minimize the process and improve productivity. This tool balances the cycle time, manpower, and TAKT time [30]. It helps to design an improved facility layout that optimizes the process by minimizing cycle and lead times and improves overall production activities. A pull system must be applied in complex manufacturing systems to minimize congestion in the production line. The current manufacturing system in the automotive industry relies on a push system to feed the assembly and production lines because of the schedules and expected cycle times. This approach results in having too much buffer stock and unnecessary utilization of resources. The pull system promotes lean manufacturing principles because all the required resources are pulled when necessary and required for use. This results in easily identifying waste and

unsafe practices in the complex process. Material handling and safety are critical aspects of production free flow and advance the value chain at an optimal cycle time [31]. The constraint of material is a result of unlevelled production schedules causing too many changeovers in material handling. To validate the populated and visualized concerns, the cycle times are categorized as VA and NVA to show how much the populated concern affects the process. The study of the NVA contributed to the cycle time envisaged as the situation and reckoning with the presence of one of the seven waste lean. The contribution of the VA and NVA is shown in the Yamazumi diagram in Figure 8, which depicts how the processes are responding to the TAKT time. This is based on empirical data from the observation during time measurement. The Yamazumi confirms the issues observed and the NVA recorded by showing how close the process cycle times are to the TAKT time.

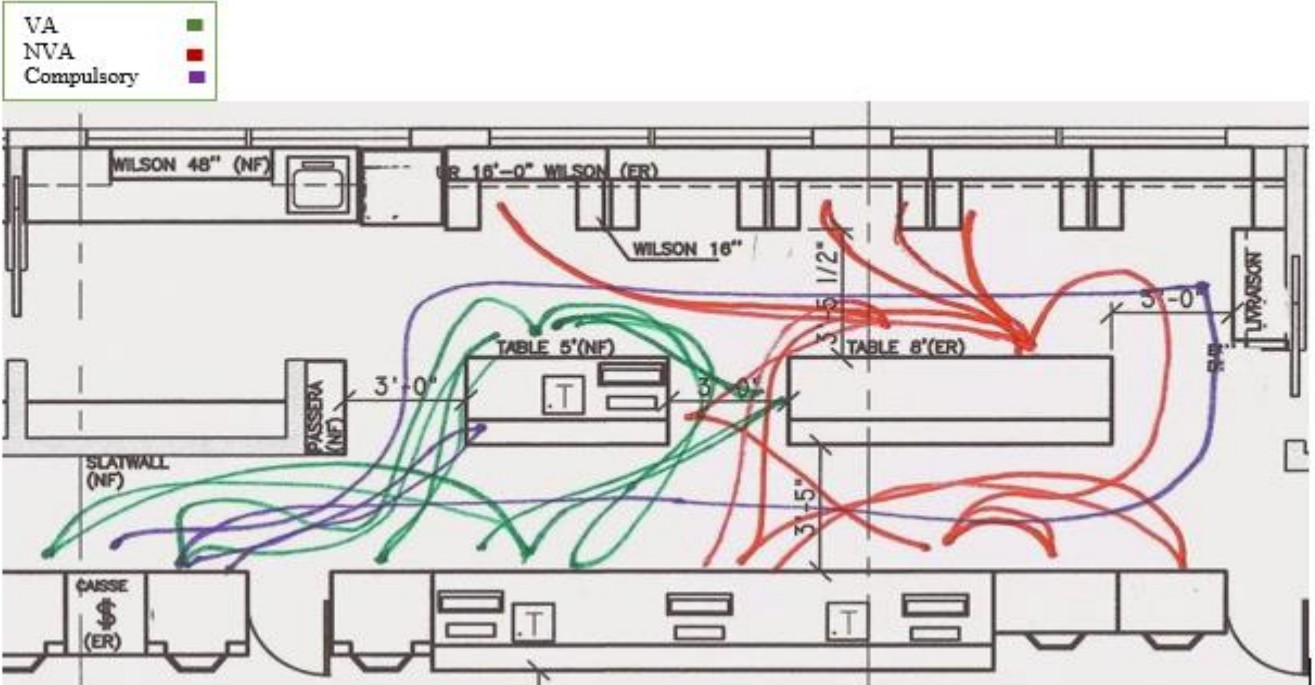

**Figure 8.** The Spaghetti diagram for visualized workstation movement, material handling, and parts storage (VA—value added times; NVA—Non-value-added times; Compulsory—operations).

## 6. Conclusions

Simulated value stream mapping proved that traditional VSM could not identify all process non-value adding by highlighting workstations that are affected by the dynamic of the production processes. These dynamics showed that in traditional VSM, a drifting operation is not identified due to the static construction of VSM. It is one of the findings identified that some of the processes in the value chain are automatic constraints and cannot be identified using traditional VSM. This shows that SVSM provides more production process visualization, which is important in every project before implementation. This gives necessary information to make decisions on the feasibility of the improvements and their impact on the entire production system.

This study identified stations with short cycle times that cause bottlenecks in the station that follows and disturbs the free flow. Hence, in other production systems, there are subassembly stations to avoid these unnecessary bottlenecks. This is one of the concerns of South African automotive SMEs. Processes were established and aligned to serve the demand at the time; changes in the global market do not change how things are done. This is one of the findings shown by the painting and drying station; removing these two stations provides production free flow and improves productivity. These findings emphasized that Kaizen projects undertaken by SMEs must align with customer KPIs where optimized

processes are crucial to productivity improvement. SMEs need to have a thorough approach to optimize the use of space to avoid excessive movement that will make the process take more time than expected. Optimized processes support quality and can support any change in the production system. Cycle and lead time are the key performance indicators of any production system; they determine how well each part of the system is doing. This research study shows a need for continuous process improvement to reduce the cycle time to be productive and competitive. As seen in what VSM displayed relative to the process cycle time and what was identified as NVA, failure to identify these process constraints will fail in the implementation phase of the project. Hence, this research project introduced SVSM, where an immediate trial is undertaken using the same parameters as VSM. The unforeseen NVA during the study automatically appears or is triggered by identifying delays in the processes. These are usually caused by parts presentation, unboxing, and other waste mentioned earlier in the study. Further research is on development machine learning approach to enhance the accuracy manufacturing process optimization to improve productivity.

**Author Contributions:** Conceptualization, F.P. and O.T.A.; methodology, F.P., software, O.T.A.; validation, F.P., O.T.A. and K.M.; formal analysis, F.P. and O.T.A.; investigation, O.T.A.; resources, F.P. and O.T.A.; data O.T.A.; editing, K.M.; visualization, O.T.A.; supervision, O.T.A. and K.M.; project administration, F.P., funding acquisition, K.M. All authors have read and agreed to the published version of the manuscript.

**Funding:** This research was funded by National Research Foundation (NRF), grant number 123575, and the APC was funded by the Research Chair in Future Transport Manufacturing Technologies.

**Data Availability Statement:** Not applicable.

**Acknowledgments:** The researchers acknowledge the support and assistance of the Industrial Engineering Department of Tshwane University of Technology, Gisela Rail, and the National Research Foundation (123575) of South Africa for their financial and material assistance in executing this research project. The opinions presented in this paper are those of the authors and not the funders.

**Conflicts of Interest:** The authors declare no conflict of interest.

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
