# Peer review of "Productivity Improvement Using Simulated Value Stream Mapping: A Case Study of the Truck Manufacturing Industry"

_processes, doi:10.3390/pr10091884_

Round 1
Reviewer 1 Report
The article is poorly organized.
Section 3.1.1. Current state VSM on page 4, has the same numbering as 3.1. Distribution Factor to the Assembly Line on page 6.
There are no sections 4 and 4.1
Section 4.2 is repeated twice on page 7 and page 8.
There are no sections 4.2.1 and 4.2.2. Section 4.2.3 appears directly.
There is no Figure 1 in the article. The figures are misidentified.
Article writing is not professional.
Under these conditions, it is not possible to review the content of this document.
Author Response
Response to Reviewer 1 Comments
Point 1: The article is poorly organized.
Response 1: We have improved on the organisation of the paper.
Point 2: Section 3.1.1. The current state VSM on page 4, has the same numbering as 3.1. Distribution Factor to the Assembly Line on page 6.
Response 2: It has been corrected
Point 3: There are no sections 4 and 4.1
Response 3: It has been incorporated as 4. 4.1 and 4.2
Point 4: Section 4.2 is repeated twice on page 7 and page 8.
Response 4: Corrected
Point 5: There are no sections 4.2.1 and 4.2.2. Section 4.2.3 appears directly.
Response 5: Corrected
Point 6: There is no Figure 1 in the article. The figures are misidentified.
Response 6: Figure 1 has been identified correctly
Point 7: Article writing is not professional.
Response 7: This has been adjusted in adherence to the Processes Journal guidelines
Point 8: Under these conditions, it is not possible to review the content of this document.
Response 8: We acknowledged the efforts of the reviewer on the listed comments and necessary adjustment has been made to improve of the journal write up professionally

Reviewer 2 Report
Dear Authors,
Thank you for compiling the article, which, however, needs significant improvement.
The language is incomprehensible in several places, such as:
63: The methodology in section 3 discusses how the type of the applied data is ideal for the study.
78: VSM is a tool that enhances the use of other waste reduction techniques to enhance core functionality support using VSM in waste reduction and proper strategies on how to reduce the time wasted on the system.
Some statements in the literature review are not legitimate (or cannot be derived from the cited articles), such as:
48: Simulation has been around for decades but has not been taken as an effective tool in the production and manufacturing environment because of its stochastic nature - This is true for DES only
112: Lack of automation (process variability) in SME operations and for the entity to become accustomed to a high demand that requires complex structures in their production system has become a challenge. This makes it hard to complete a feedback loop to analyze the system behavior and enable performance improvement in a manufacturing system. The application of simulation models can minimize these challenges by analyzing these complex structures [17]. (These complex structures are feedback loops, which are simulated in SD method).
There are repetitions, similar and identical fragments in the article, such as:
225: The work measurement technique used in this study is determined by the time series involved in completing one unit in the production line and to compare different cycle times the determination of the value flow and reliability of the process of the production process.
241: Work measurement was used to compare different cycle times in the production process, and determine the value flow and the reliability of the process.
346: Value stream mapping is constructed using the cycle time, change over time, operators, and process description. This process visualizes the production system and the behavior of each workstation. This populates the waste recorded while collecting data during observation of the current state of VSM. All elements are exhibited to point out which areas have problems and potential solutions are developed based on what is visible. To validate the populated and visualized concerns, the cycle times are categorized as VA and NVA to show how much the populated concern affects the process.
363: Value stream mapping is constructed using the cycle time, change over time, operators, and process description. This process visualises the production system and the behaviour of each workstation. This populates the waste recorded while collecting data during observation of the current state of VSM. All elements are exhibited to point out which areas have a problem and potential solutions are developed based on what is visible. To validate the populated and visualized concerns, the cycle times are categorized as VA and NVA to show how much the populated concern affects the process.
Please note that only selected examples of shortcomings in the article were mentioned.
While I appreciate the work you have put into conducting the research, I suggest you revise the article and resubmit.
Author Response
Response to Reviewer 2 Comments
Point 1: The language is incomprehensible in several places, such as:
63: The methodology in section 3 discusses how the type of the applied data is ideal for the study.
Response: The methodology in section 3 discusses how conventional value stream mapping (VSM) is complemented by simulation where applied data is ideal to describe the evidence of complementary time study data method, and data analysis by VSM and SVSM schematic diagrams. The data were simulated using AnyLogic Discrete Event Simulation (DES) and manual VSM parameters.
78: VSM is a tool that enhances the use of other waste reduction techniques to enhance core functionality support using VSM in waste reduction and proper strategies on how to reduce the time wasted on the system.
Response: VSM is a tool that can easily be used with other waste reduction techniques to enhance core functionality that supports using VSM as a waste reduction tool and proper strategies on how to reduce the time wasted on the system
Point 2: Some statements in the literature review are not legitimate (or cannot be derived from the cited articles), such as:
48: Simulation has been around for decades but has not been taken as an effective tool in the production and manufacturing environment because of its stochastic nature - This is true for DES only
Response: deleted
112: Lack of automation (process variability) in SME operations and for the entity to become accustomed to a high demand that requires complex structures in their production system has become a challenge. This makes it hard to complete a feedback loop to analyze the system behavior and enable performance improvement in a manufacturing system. The application of simulation models can minimize these challenges by analyzing these complex structures [17]. (These complex structures are feedback loops, which are simulated in SD method).
Response: Deleted
112: Lack of automation (process variability) in SME operations and for the entity to become accustomed to a high demand that requires complex structures in their production system has become a challenge. ( Influences of the Industry 4.0 Revolution on the
Response: The discussion has been improved.
Point 3: There are repetitions, similar and identical fragments in the article, such as:
225: The work measurement technique used in this study is determined by the time series involved in completing one unit in the production line and to compare different cycle times the determination of the value flow and reliability of the process of the production process.
Response:
241: Work measurement was used to compare different cycle times in the production process, and determine the value flow and the reliability of the process.
Response: Deleted
346: Value stream mapping is constructed using the cycle time, change over time, operators, and process description. This process visualizes the production system and the behavior of each workstation. This populates the waste recorded while collecting data during observation of the current state of VSM. All elements are exhibited to point out which areas have problems and potential solutions are developed based on what is visible. To validate the populated and visualized concerns, the cycle times are categorized as VA and NVA to show how much the populated concern affects the process.
Response: Deleted
363: Value stream mapping is constructed using the cycle time, change over time, operators, and process description. This process visualises the production system and the behaviour of each workstation. This populates the waste recorded while collecting data during observation of the current state of VSM. All elements are exhibited to point out which areas have a problem and potential solutions are developed based on what is visible. To validate the populated and visualized concerns, the cycle times are categorized as VA and NVA to show how much the populated concern affects the process.
Response: Deleted
Please note that only selected examples of shortcomings in the article were mentioned.
While I appreciate the work, you have put into conducting the research, I suggest you revise the article and resubmit it.
Response: The article has been revised and presented in the corrected form for further consideration.

Reviewer 3 Report
Manuscript ID processes - 1881740
Title: «Productivity improvement using simulated value stream mapping: A case study of the truck manufacturing industry».
Presented study opens up defined prospects in this field of knowledge. Overall, the topic of this study is relevant, and the manuscript was not bad organized and written. Manuscript entitled "Productivity improvement using simulated value stream mapping: A case study of the truck manufacturing industry " of interest to a highly ranked journal like "Processes".
This work includes a not quite adequate structure:
* Introduction (p. 1 – 2);
* Literature Review (p. 2 – 4);
* Materials and Methods (p. 4 – 16);
* Conclusions (p. 16 – 17).
Dear authors, the manuscript is informative but it has some inadequacies. I hope that next suggestions can help to improve the manuscript.
1. In the Abstract, please make clearer the aim of the study. In my opinion, general aim of the study is not formulated.
2. To increase the visibility of article in the future, it is necessary to adjust the keywords.
3. Regarding the Introduction section. At the end of the section, I would suggest that the authors add a conceptual model or study framework for a better perception of article by readers.
4. Attentive text proofreading is needed. Section and chapter numbering is broken. Where is RESULTS section?
5. Indicate the directions of further research or improvements.
Author Response
Response to Reviewer 3 Comments
Point 1: This work includes a not quite adequate structure:
* Introduction (p. 1 – 2);
* Literature Review (p. 2 – 4);
* Materials and Methods (p. 4 – 16);
* Conclusions (p. 16 – 17).
Response 1: We acknowledged the comments and appreciate the reviewer time to dissect the paper. We have adjusted the structure and reflect the comments to the include the listed points.
Point 2: In the Abstract, please make clearer the aim of the study. In my opinion, general aim of the study is not formulated.
Response 2: The abstract has been rephrased with the aim in bolded.
The accumulation of process waste in the production line causes fluctuations, bottlenecks, and increased inventory in workstations disrupting process flow. In this paper, the optimal process flow that will improve productivity using simulated value stream mapping (SVSM) for decision-making to provide consistency, minimise errors and non-value adding times in the implementation phase of VSM in the truck manufacturing industry. The proposed methodology applied discrete event simulation for production process operations improve-ment to eliminate non-value adding times and provides good quality products at the lowest cost and highest efficiency. The results are the analysis of the current state of the production system in a South African truck manufacturing industry as a potential solution for the pro-duction system's future state. The identified non-value adding times in the 6 most critical workstations were eliminated by SVSM resulting in a productivity improvement of 4%, most importantly bringing the productivity to 95% and total cycle time improvement to 451 for small units and 466 for large units. The results proposed combined VSM and Simulation techniques based on empirical data from the observation during time measurement. The Yamazumi confirms the issues observed and the NVA recorded by showing how close the process cycle times are to the TAKT time. which enhance the LEAN application by DES to increase productivity and performance improvement to remain competitive in the global economy. which enhance the LEAN application by DES to increase productivity and performance improvement to remain competitive in the global economy.
Point 3: To increase the visibility of article in the future, it is necessary to adjust the keywords.
Response 3: Simulated value stream mapping; productivity improvement; Lean enterprise; cyber-physical systems; Spaghetti/Yamazumi diagram;
Point 4: Regarding the Introduction section. At the end of the section, I would suggest that the authors add a conceptual model or study framework for a better perception of article by readers.
Response 4: Section 4 presents the results that show the study application was to improve the productivity of the truck manufacturing industry, while the last section presents the conclusion based on empirical data from the observation during time measurement. The Yamazumi confirms the issues observed and the NVA recorded by showing how close the process cycle times are to the TAKT time. which enhance the LEAN application by DES to increase productivity and performance improvement to remain competitive in the global economy.
Point 5: Attentive text proofreading is needed. Section and chapter numbering is broken. Where is RESULTS section?
Response 5: Text proofreading has been done and chapter numbering has been corrected. The results are presented in section 4
Point 6: Indicate the directions of further research or improvements.
Response 6: Further research is on development machine learning approach to enhance the accuracy manufacturing process optimization to improve productivity.

Round 2
Reviewer 1 Report
The article has been improved and meets the standards for publication.
Reviewer 2 Report
I accept the article in revised form and recommend for publication.